# Mitogen-Activated Protein Kinase (MAPK) and Obesity-Related Cancer

**DOI:** 10.3390/ijms21041241

**Published:** 2020-02-13

**Authors:** Fionán Donohoe, Michael Wilkinson, Eva Baxter, Donal J. Brennan

**Affiliations:** 1Ireland East Hospital Gynaeoncology Group, UCD School of Medicine, Mater Misericordiae University, D07R2WY Dublin 7, Ireland; fionandonohoe@gmail.com (F.D.); michael.wilkinson87@gmail.com (M.W.); 2Queensland Centre for Gynaecological Cancer Research, The University of Queensland, Brisbane QLD 4029, Australia; e.baxter@uq.edu.au; 3Systems Biology Ireland, UCD School of Medicine, Belfield, D04V1W8 Dublin 4, Ireland

**Keywords:** obesity, MAPK, cancer, endometrial cancer, hepatocellular cancer

## Abstract

Obesity is a major public health concern worldwide. The increased risk of certain types of cancer is now an established deleterious consequence of obesity, although the molecular mechanisms of this are not completely understood. In this review, we aim to explore the links between MAPK signalling and obesity-related cancer. We focus mostly on p38 and JNK MAPK, as the role of ERK remains unclear. These links are seen through the implication of MAPK in obesity-related immune paralysis as well as through effects on the endoplasmic reticulum stress response and activation of aromatase. By way of example, we highlight areas of interest and possibilities for future research in endometrioid endometrial cancer and hepatocellular carcinoma associated with non-alcoholic fatty liver disease (NAFLD), non-alcoholic steatohepatitis (NASH) and MAPK.

## 1. Introduction

Obesity is a major public health concern that develops as a result of chronic caloric excess. The excess energy is stored as lipid in adipose tissue and may accumulate in other metabolic organs (such as the liver) and skeletal muscle. Increased lipid accumulation significantly alters the normal metabolic milieu and creates an environment that chronically transmits a signal of nutrient excess to the cell, with multiple consequences, many of which are poorly understood.

Global obesity rates have been increasing dramatically over the last number of years, with rates now almost triple what they were in 1975. According to the World Health Organisation, there were 1.9 billion overweight adults worldwide in 2016, with 650 million of those classified as obese, representing 13% of the world’s population. Once considered a problem confined to the developed world, obesity rates are also increasing in middle- and low-income countries with a fifty percent increase in the numbers of overweight children in Africa since the year 2000 [1,2].

The effect of obesity on cardiovascular disease and diabetes is well established, however the association between obesity and increased risk of certain types of cancer is a relatively new observation and the underlying mechanisms have not been fully elucidated. Improved understanding of the molecular interactions between obesity and cancer is important if any future public health interventions are to be successful in reducing population obesity levels and the associated cancer risk.

MAPK pathways are signalling modules that transduce extracellular and intracellular signals to regulatory networks within the cell via the phosphorylation of key protein targets. Each cascade is activated by extracellular signals which lead to the activation of the MAPK through successive activations of MAPK kinase kinase (MAPKKK) and MAPK kinase (MAPKK). There are three main families of MAPK—the extracellular signal regulated kinases (ERKs), the Jun amino terminal kinases (JNKs) and the stress activated protein kinases (p38/SAPKs). The ERKs respond primarily to growth factors and mitogens to induce cell growth and differentiation, while the JNKs respond primarily to stress such as ionising radiation and oxidative stress and are involved in apoptosis, cytokine production, inflammation and metabolism. Similarly, the p38 MAPKs are strongly activated by cytokines and cellular stress. The activation of the p38 MAPK pathway contributes to inflammation, apoptosis, cell differentiation and cell cycle regulation [3]. The involvement of the ERK pathway in obesity-related cancer is poorly understood at present and as the JNKs and p38 are thought to be the more important groups influencing obesity, they will be the focus of this review.

In this review, we discuss the relationship between obesity and cancer and the mechanisms underlying obesity-related carcinogenesis, with a focus on mitogen-activated protein kinase (MAPK) signalling. To do this, we have chosen endometrial cancer as a case study, for the following reasons, (a) the epidemiological association between obesity and endometrial cancer is irrefutable, and (b) weight loss reduces endometrial cancer. We also discuss the role of obesity and MAPK in hepatic steatosis.

## 2. Obesity-Related Cancers—Epidemiological Perspective

Epidemiological evidence linking obesity to cancer development is consistent and compelling for cancer of the oesophagus (adenocarcinoma), kidney, breast (postmenopausal), colon, rectum, endometrium, liver, thyroid, gallbladder, ovary and pancreas [2,4]; however, the relative risks differ between cancer types. The most compelling relationship with obesity has been demonstrated in endometrioid endometrial cancer, adenocarcinoma of the oesophagus and cholangiocarcinoma. 

## 3. Obesity and Metabolic Dysfunction as Cancer Risk Factors

Overall there is a modest but important increase in the relative risk of developing any cancer in the order of 1.06–1.62 for each increment of body mass index BMI of 5 kg/m^2^ [4]. In 2012, it was estimated that 3.6% of all new adult cancers over 30 years of age were attributable to high BMI, with the impact greater in countries where obesity was more prevalent, and a greater burden of risk for women than for men [5]. In those with a BMI greater than 40 kg/m^2^, the risk of death attributable to cancer is higher than those with a normal BMI, although most of the observed decrease in life expectancy in this group is attributable to cardiovascular events [6], again highlighting the importance of public health interventions in this population. Obviously, there are multiple confounding factors, such as levels of physical activity, energy balance and metabolism in obesity-related carcinogenesis, and many have not been fully explored from an epidemiological perspective [7]. 

Of these confounders, metabolic dysfunction may be the most important confounder. The concept of healthy obese patients, i.e., those with no evidence of diabetes, hypertension or other co-morbidities, has gained a lot of traction in recent years. As a result, there has been an increased interest in the role of metabolic aberrations in cancer risk and outcome. A number of studies have demonstrated that individuals with metabolic syndrome and diabetes have an increased risk of cancer of the pancreas, rectum and endometrium [8,9,10]. In this setting, elevated blood glucose has been associated with increased cancer risk in a number of prospective studies. Diabetes is now a recognized risk factor for cancer [8,11,12,13,14,15]. A large collaborative study of over 550,000 patients in six prospective cohort studies as part of the Metabolic Syndrome and Cancer Project (Me-Can) demonstrated that abnormal glucose metabolism—independent of BMI—was associated with increased cancer risk at various sites in both men and women [16]. A follow-up study from the same collaborative group demonstrated that each standard deviation increase in a metabolic risk score was associated with 56% increased risk in endometrial adenocarcinoma. Within this cohort, the association between BMI and cancer incidence was much weaker than the association between serum glucose levels and cancer incidence [17]. Furthermore, BMI was not associated with cancer mortality, while elevated blood glucose was a significant predictor of cancer mortality. 

## 4. Impact of Weight Loss on Cancer Risk

The epidemiological links between obesity, metabolic dysfunction and cancer development are clear. What, then, is the impact of weight loss in terms of cancer risk? Observational studies have shown a decrease in cancer-related mortality for those who lose weight, to a larger extent for women than for men [18,19]. Larger cohort studies demonstrate that metabolic surgery decreases cancer incidence and mortality when compared to obese controls [20]. The Swedish Obese Subjects (SOS) study demonstrates decreased incidence of cancer in women who underwent metabolic surgery when compared with biologically matched controls [21] a finding reciprocated in a larger American cohort which demonstrated similar results, with a marked decreased risk for obesity-related cancers [22]. 

A recent small (*n* = 6) prospective cohort study demonstrated the ability of metabolic surgery to reverse complex atypical hyperplasia of the endometrium, a known endometrial cancer precursor. Hyperplasia resolved in the majority of cases (5/6; 83%) with 3/6 (50%) resolving with metabolic surgery alone [23]. Endometrial biopsies taken before and after surgery demonstrated reduced expression of oestrogen and progesterone receptors at 2 and 12 months after metabolic surgery and a 17% reduction in the Ki-67 proliferation index seen 12 months after metabolic surgery. This was associated with a significant reduction in pAKT (a serine threonine kinase also called protein kinase B) and increased expression of PTEN (phosphatase and tensin homolog). No changes were seen in pERK, suggesting that the impact of weight loss on obesity-related carcinogenesis in the endometrium may not be dependent on classical MAPK signalling in the endometrium. However, there is evidence to suggest that MAPK signalling is reduced in endometrioid endometrial tumours in obese women compared with non-obese—making the MAPK pathway a potential target for endometrial cancer in non-obese women [24]. As expected, and based on previous metabolic surgical cohorts, circulating biomarkers for both insulin resistance (Haemoglobin A1C, HOMA-IR (homeostatic model assessment of insulin resistance)) and inflammation (C-Reactive Protein (CRP), Interleukin-6 (IL6)) were significantly reduced after metabolic surgery. 

The small study by MacKintosh et al. [23], demonstrates the importance of longitudinal studies to assess the impact of weight loss on established [25] and emerging [26] hallmarks of cancer development. This includes the pathways involved in inflammation, angiogenesis, invasion and metastasis, cellular energetics and genomic instability.

## 5. Obesity-Related Hyperinsulinaemia

Obesity is a toxic state of increased circulating levels of oestrogen, hyperinsulinemia and persistent low-grade inflammation. Obesity-related insulin resistance leads to the increased secretion of insulin, which in turn leads to a state of hyperinsulinemia. In terms of its signalling, insulin binds to either insulin receptor A or B and exerts its metabolic effects via various pathways both upstream and downstream of Akt. Its mitogenic effects are transmitted through the ERK MAPK pathway which are implicated in cancer development and progression [27]. Hyperinsulinemia is also associated with increased levels of insulin like growth factors (IGF) 1 and 2, in part due to the decreased synthesis of IGF binding proteins (IGFBP), which serve to bind and thus inactivate IGF1 and 2 [28]. IGF 1 and 2 bind the IGF receptor or a receptor hybrid of the insulin receptor and the IGF receptor and go on to affect cell survival, proliferation, growth and differentiation [29,30].

The PI3K-Akt-mTOR pathway is also intimately linked to oestrogen-mediated endometrial proliferation. There are multiple signalling pathways upstream from PI3K/Akt/mTOR that also play key roles in regulating cellular proliferation; however, in obesity-related cancers, insulin receptors and insulin-like growth factors (IGF) play a key role. The IGF1/PI3K-Akt-mTOR pathway plays an important role in the regulation of cellular growth, and the proliferation and differentiation of cells in multiple organs [31,32]. Binding of IGF-1 to its receptor leads to the tyrosine kinase-induced phosphorylation of the PI3K-Akt-mTOR pathway and the subsequent enacting of its downstream effects on cellular growth. Several studies have implicated the IGF1 receptor in both obesity-related and non-obesity related endometrial cancer [33,34,35]. Directly, IGF1 and its receptor increase the activity of the PI3K-Akt-mTOR pathway, thus increasing cellular proliferation and survival. Indirectly, the increased insulin levels inherent to obesity lead to increased circulating oestrogen and the ensuing decrease in IGFBP1, a negative regulator of IGF1. The resultant IGFBP1 downregulation leads to a major increase in IGF1 bioavailability and further upregulation of PI3K-Akt-mTOR pathway [36]. The relationship between IGF and other circulating growth factors is well understood and has been reciprocated in other organs, such as the liver. Furthermore, aberrant p38 activity has been shown to be potentially implicated in insulin resistance [37,38]. Next, we will discuss the tumour microenvironment and its role in obesity-related carcinogenesis.

## 6. Obesity-Related Inflammation

Obesity is a state of chronic inflammation. This inflammation has been implicated in the development of cancer in obese individuals [39]. Use of non-steroidal anti-inflammatory drugs (NSAIDs) has been associated with a reduced risk of endometrial cancer, particularly in obese women [40,41,42], implying a causative role for inflammation in obesity-related endometrial cancer.

Adipose tissue secretes several important cytokines involved in mediating this low-grade inflammation. The most important of these are adiponectin and leptin [43]. Adiponectin has strong inverse associations with several pro-inflammatory cytokines—namely tumour necrosis factor alpha (TNFa) and interleukin-6 (IL6) [44], which are elevated in obesity. Visceral obesity and ectopic fat production are associated with hypoadiponectinemia [45]. Leptin levels, on the other hand, are elevated in obesity and are positively correlated with levels of inflammatory cytokines, namely IL1 beta, IL6 and IL12 amongst others [46]. Interestingly, weight loss has been associated with the attenuation of this inflammatory response [47,48].

All forms of MAPK are involved in the innate immune response to pathogen infection and tissue damage in the normal, non-cancerous state. This occurs through activation of pattern recognition receptors (PRRs) on the cell surface and in the cytoplasm of immune cells. Most MAPKs are activated via Toll-like receptors (TLRs) which, through a series of downstream factors, activate MAPKKK, MAPKK and MAPK, which in turn lead to the increased production of pro-inflammatory cytokines such as TNF and IL1beta as well as recruitment of M1 macrophages [49], all of which contribute to a functional immune response.

Obesity-related immune dysfunction may limit anti-tumour immune responses. Although obesity causes a pro-inflammatory response, as evidenced by the expansion and proliferation of pro-inflammatory macrophages in adipose tissue [50,51] with an associated loss of regulatory immune cells [52,53], this does not result in anti-tumour immune response—as obesity appears to affect immune cell function. Obese individuals have been shown to have altered or impaired function of circulating natural killer cells [54,55] as well as dendritic cells [56,57].

## 7. Obesity-Related Immune Dysfunction

A clear understanding of the mechanisms underlying immune dysfunction may improve our understanding of how the obese state allows tumours to evade immune surveillance in certain cancer types. Obesity induces both cell-specific, and global transcription changes. Chief among the global transcriptomic alterations induced by obesity are changes in intrinsic metabolic pathways [58,59]. In brief, immune cells from obese individuals and mice are metabolically paralysed, meaning that they are unable to induce glycolysis or oxidative phosphorylation upon activation [60]. The dynamic ability of immune cells to switch from a quiescent to metabolically active state is essential for anti-tumour immune response. The negative impact of obesity on several immune cell populations, including invariant Natural Killer (NK) T cells, NK cells, dendritic cells, and mucosal-associated invariant T (MAIT) cells is now well established [53,54,61,62].

Recent elegant studies have highlighted the crucial role of intrinsic cellular metabolism in driving immune cell functions [63,64]. These studies demonstrated that the manipulation of immune cell metabolism by inducing or preventing glycolysis using synthetic compounds can dramatically alter cytokine production, effector function and memory formation. More importantly, it has been consistently demonstrated in multiple types of immune cells that obesity alters immunometabolism and thus impairs their ability to mount an anti-tumour immune response.

The interaction between immunometabolism and MAPK signalling in tumour infiltrating leukocytes has not been investigated in detail, however monocyte adhesion and activation are dependent on glycolysis, which causes the activation of p38 MAPK [65]. Therefore, obesity-related immune paralysis may play an important role in the loss of the tumour suppressive effects of p38 MAPK signalling.

The role of MAPK signalling in immune cells is further highlighted by recent findings in the MAIT cells. These are a population of non-MHC-restricted T cells which are responsible for the response to viral and bacterial infection through rapid production of various cytokines [66], whose effector function is reduced in obese humans [62]. MAIT cells upregulate their rates of glycolysis to execute this function; however, in obese adults the rate of glycolysis in MAIT cells is not upregulated when compared with MAIT cells in lean adults, and this is related to impaired function of the mTOR complex 1 (mTORC1)—composed of mTOR, Raptor, and GβL—and DEPTOR (DEP domain containing MTOR interacting protein) appears to be a key regulator of this response [63].

The mTORC1 signalling complex acts as both a nutrient sensor and a metabolic regulator and is intrinsically linked to glycolytic metabolism in effector lymphocytes [67,68]. mTORC1 signalling in immune cells is reduced in obesity; however, a recent study of The Cancer Genome Atlas (TCGA) PanCancer atlas identified a novel activator of mTORC1—MAPK4 (ERK4), an atypical MAPK that lacks the canonical Thr-X-Tyr (TxY) activation motif for phosphorylation by the dual Ser/Thr and Tyr MAPK kinase (MAPKK) [69,70], which was associated with poor prognosis in a panel of lung, bladder and central nervous system (CNS) cancers, none of which are obesity-related [71]. MAPK4 caused the activation of AKT independent of PI3K and PTEN, and levels of MAPK4 expression in obesity-related cancers such as endometrial and cholangiocarcinoma were extremely low, suggesting that this interaction requires further investigation.

Further links between immune cells, obesity and MAPK have been demonstrated through examining the effects of leptin in T helper immune cells, albeit in allergic airways disease. Leptin activated mTORC1 and induced ERK and p38 MAPK phosphorylation in TH1 and TH2 cells, and both MEK and mTOR inhibitors blocked its effect on T helper cell proliferation, survival and cytokine production [72,73]. Taken together, these data demonstrate the important relationship between obesity and immunometabolism, MAPK signalling and effector function and highlight the need for further research in this area.

## 8. The Gut–Brain Axis in Cancer

In addition to the evolving obesity-immunometabolism story, the role of the gut-brain axis has received a lot of attention in the obesity literature but has been relatively under-investigated in the cancer literature. The gut–brain axis consists of multidirectional communication between the enteric nervous system (ENS) and the central nervous system (CNS), linking reward centres of the brain with peripheral intestinal functions and is dependent on multiple endocrine and paracrine processes [74]. Animal studies suggest that absorbed gut luminal factors (such as nutrients) and circulating gut hormones directly activate the arcuate nucleus of the hypothalamus (ARC). The ARC contains two groups of antagonistic neurons. The first synthesize pro-opiomelanocortin (POMC)-derived peptides and CART (cocaine- and amphetamine-regulated transcript) peptides, which reduce food intake, and the second synthesize NPY (neuropeptide Y), which directly stimulates food intake. Serum CART levels are increased in obesity and fall following metabolic surgery [75,76]. We have previously demonstrated that CART causes ligand-independent activation of oestrogen receptors in breast cancer cell lines via activation of ERK-MAPK signalling. Increased CART expression was associated with poor prognosis in lymph node negative, ER positive post-menopausal breast cancer—a well-recognised obesity related disease. These data demonstrate the interaction between circulating factors and intracellular signalling in obesity-related cancer [77].

The intestinal microbiome is another important growth area highlighting the relationship between obesity, cancer and the MAPK pathway. The composition of the gut microbiome varies between individuals and in response to dietary intake. It lives in a mutualistic relationship with the host and is a key contributor to the hosts metabolism through the creation of vitamins and other substrates required for its physiology. Changes in the composition and function of the microbiota—referred to as gut microbiota dysbiosis—have been implicated in human diseases such as gastrointestinal disorders, cardiovascular disease, neurological disease and cancer [78]. Obesity is known to be associated with gut microbiota dysbiosis, although consistent alterations specific to obese humans have not yet been described [79]. Dual specificity protein 6 (DUSP6) is a MAPK phosphatase (MKP) also known as MKP3 which specifically inactivates ERK. Mice deficient in this enzyme are resistant to diet-induced obesity [80]. These mice also have a unique gut microbiome which, when transferred to wild type germ free mice, increased energy expenditure and reduced weight gain. NOD-like receptor protein 12 (NLRP12) is a potent mitigator of inflammation and inhibits ERK activation. Mice deficient in NDLRP12 have increased weight gain, adipose tissue inflammation and MAPK activity [81]. This mechanism is dependent on changes in the gut microbiome of the mice. These studies show that microbiota homeostasis is potentially dependent on MAPK pathways.

The microbiome is also involved in changes in gut inflammation, and again the MAPK pathway is potentially involved here with dietary-induced changes in the gut affecting T cell proliferation and differentiation via the p38 and JNK pathways [82]. This shows that the microbiota potentially affects intestinal immune function via the MAPK pathway.

## 9. Obesity-Related Endocrine Abnormalities

Hormonal influences also play an important role in the pro-neoplastic extra-cellular milieu associated with obesity. A report from the International Agency for Research on Cancer concluded that there was strong evidence for chronic inflammation and sex hormone metabolism mediating the relationship between obesity and cancer, whilst evidence for insulin and IGF signalling was moderate [83].

Adipose tissue is a source of adipokines, inflammatory cytokines and hormones which can promote tumour development or growth. Amongst other actions, inflammation increases local expression and activity of aromatase [84,85], the key enzyme catalysing oestrogen synthesis. In postmenopausal women, adipose tissue is one of the few sources of oestrogen which has been recognised as a risk factor for endometrial cancer for many decades [86]. However, the association between obesity and elevated levels of oestrogen remains disputed. Numerous studies have reported that circulating levels of oestrogens are significantly higher in postmenopausal women who are obese than in non-obese women [87,88,89,90,91,92,93,94,95]; however, others have found no significant association [96,97,98]. Aromatase inhibitors have been shown to reduce oestrogen levels in postmenopausal women with breast cancer, though obese women still had more recurrences and poorer prognoses than non-obese women [98,99,100,101], indicating that there is more to the link between obesity and cancer than just oestrogen.

Up to 80% of endometrial and breast cancers express the oestrogen receptor ERα. The selective ERα modulator tamoxifen is a key component of treatment for ERα-positive breast cancer, however it increases the risk of endometrial cancer (reviewed in [102]), suggesting that ERα signalling varies between the two cancer types. We recently demonstrated that ERα-associated networks differ between endometrial and breast cancer and that these networks have distinct regulators, with a unique role for XBP1 (X-box binding protein 1) in endometrial cancer [103]. TCGA had previously predicted XBP1 to be an important part of a regulatory signalling hub in endometrial cancer [104] and subsequently demonstrated that XBP1 was enriched in endometrial tumours versus breast tumours [105]. We showed that oestrogen increased co-recruitment of ERα and XBP1 to target genes and that knockdown of XBP1 blunted the oestrogen-mediated induction of oestrogen-responsive genes only in endometrial cancer cells and not breast cancer cells. Furthermore, high levels of XBP1 were an independent prognostic factor for improved progression-free survival only in women with endometrial cancer, suggesting that XBP1 plays a distinct role in oestrogen signalling in endometrial cancer [103].

XBP1 is an oestrogen-responsive gene [103,106], however it is also activated in response to endoplasmic reticulum (ER) stress which is caused by numerous factors, including obesity [107]. IRE1α, an ER transmembrane protein sensor, splices and activates XBP1. Spliced XBP1 (XBP1s) translocates to the nucleus where it initiates the transcription of genes related to ER stress, as well as unrelated genes associated with adipogenesis, inflammation and lipid metabolism. Nuclear translocation of XBP1s is enhanced by p38 MAPK and PI3K and studies in mice have shown that this interaction is reduced in obesity, preventing the resolution of ER stress and disrupting metabolic homeostasis [108,109], again highlighting the recurrent theme that obesity impairs p38 MAPK function and may impact on its tumour suppressive functions.

## 10. Endoplasmic Reticulum Stress and Obesity

XBP1 is classically associated with ER stress. The endoplasmic reticulum is an intracellular organelle with two constituent parts—the smooth ER which has no ribosomes attached and the rough ER which has abundant ribosomes present [110]. Rough ER is involved in membrane protein synthesis and tracking while smooth ER is involved in lipid metabolism and calcium homeostasis [111,112]. Within the ER, both rough and smooth, there are various complex subregions which support a variety of pathways involved in cell homeostasis and survival [111]. Obviously then, adaptation of the ER to various stresses is essential for cell function and survival [113]. The unfolded protein response (UPR) is the ER’s response to these stresses which can include inflammation as well as metabolic stress. The UPR is initiated by 3 ER membrane proteins—PERK (PKR-like eukaryotic initiation factor 2a [eIF2a] kinase), IRE1 (inositol-requiring enzyme 1), and ATF6 (activating transcription factor 6). Working together, these three membrane proteins serve to regulate protein synthesis, facilitate protein breakdown and synthesise the molecules required to restore equilibrium within the cell [114,115,116]. ER stress is caused by a wide range of cellular conditions, including but not limited to nutrient deprivation, viral infections, lipids and increased synthesis of secretory proteins [117,118]. Similar to increases in inflammation noted in the obese state, ER stress is also increased in obesity [107]. It is thought that this is due to increased synthetic demands in secretory organs, lipid accumulation, mechanical stress and nutrient availability [107].

The UPR is also active in many tumours. ATF6 and other upstream factors involved in ER stress signalling such as X-box-binding-protein 1 (XBP1), as well as targets of the response signalling pathway such as BiP (an ER chaperone), C/EBP homologous protein (CHOP) and others have been shown to be upregulated in cells of breast cancers [119], hepatocellular carcinomas [120] and oesophageal adenocarcinomas [121]—all of which are associated with obesity.

All three MAPK pathways respond to ER stress, each with different pathways and outcomes [122,123]. IRE1 activates apoptosis signal-regulating kinase 1 (ASK1), a MAPKKK which activates JNK to promote apoptosis in several cell lines [124]. Conversely, ER stress-induced activity of c-Jun, a downstream target of the JNK pathway, can reduce ER stress-induced cell death through transcription of ADAPT78 which inhibits calcineurin mediated ER stress related apoptosis [125,126]. ER stress also activates the ERK pathways to promote cell survival, although the mechanisms are not clearly understood [127].

CHOP links MAPK and ER stress through p38 MAPK-dependent phosphorylation. This pathway induces transcriptional changes in certain cell lines although the exact genes involved are as yet unknown [128]. JNK MAPK can also be activated via CHOP signalling through effects on intracellular calcium metabolism in response to ER stress, leading to increased expression of death receptors and thus apoptosis [129,130,131]. AFT6 signalling also involves p38 MAPK activity whereby sustained p38 activity leads to increased phosphorylation of AFT6 and increased transcriptional activity [132].

There are links between ER stress and MAPK signalling in specific diseases also. In melanoma cells, hepatocellular carcinoma cells [133] and breast cancer cell lines, ERK activation leads to protection from ER stress-related apoptosis, while in neuroblastoma [134] and colorectal cancer [135] cells, ERK signalling leads to increased ER stress-induced cell death.

## 11. The role of MAPK in Hepatic Steatosis

Obesity is a dominant risk factor for development of non-alcoholic fatty liver disease (NAFLD), and the increase in obesity is linked to the rising rates of both NAFLD and non-alcoholic steatohepatitis (NASH) globally [136]. Obesity also increases risk of hepatocellular carcinoma (HCC) 4.5-fold [137]. Both obesity and NAFLD are associated with insulin resistance and dyslipidaemia and ER stress is a key molecular process underlying NASH and subsequent HCC development [138]. Furthermore, metabolic surgery can facilitate up to 85% resolution of NASH, at a pre-cirrhotic level thus protecting against the development of obesity-related HCC [139,140]. We therefore sought to review the role of the various components of the MAPK in the liver and their interaction with obesity.

## 12. p38 MAPK Pathway in Hepatic Steatosis

There are four isoforms of p38 MAPK—p38 alpha, beta, delta and gamma—each of which is expressed differently in individual cells and has varied and often opposite effects even within the same cell on the same substrate [141]. P38 MAPK has myriad effects in signalling in inflammatory diseases, but with regard to cancer there is some evidence to suggest it has tumour suppressive activity [142,143]. Its suppressive activity can be largely attributed to the inhibitory effects of the alpha and beta isoforms on cell cycle checkpoint controls in different phases of the cell cycle leading to growth arrest and induction of apoptosis or cellular senescence [144,145,146,147,148]. The over-expression of p38 MAPK in cancer cells leads to suppressed proliferation and increased terminal differentiation [149]. Furthermore, p38 MAPK has also been shown to suppress tumorigenesis through effects on cell migration and implantation [150,151,152].

The p38γ and p38δ isoforms are the only isoforms to have been shown to be elevated in obese people with NAFLD when compared to non-obese people without NAFLD [153]. Furthermore, in the non-obese (BMI < 35 kg/m^2^) population those with hepatic steatosis demonstrated higher expression of these isoforms of p38 MAPK when compared with those who did not have liver disease. A lack of these isoforms has been found to be protective against steatosis in animal models [153]. Mice known to be deficient in both enzymes were protected from diet-induced hepatic steatosis and, therefore, had a lower risk of HCC. This effect was mediated through myeloid cells which were deficient in these enzymes, although a greater protection was noted with loss of p38δ than p38γ. It is thought that the deficiency of p38γ and p38δ prevented infiltration of the liver with the myeloid cells, thus reducing inflammation, and, therefore, diet-induced steatosis. This important study again highlights a theme running through this review where MAPK activity in the immune cells may be the key to understanding the role of MAPK signalling in obesity-related cancers.

MAPK phosphatases are a group of 10 catalytically active enzymes which are involved in dephosphorylation and inactivation of MAPK isoforms in mammalian cells and tissues. MKP1—the first of these enzymes to be discovered—antagonises p38 MAPK and JNK. Diet-induced obesity increased the hepatic expression of MKP1 in mice that developed steatosis. Liver-specific deficiency of MKP1 results in increased p38 and JNK MAPK activity in mouse models [154]. This is also seen in MKP5 deficient mice [155]. The role of p38 MAPK signalling in regulating metabolic responses to stress in the liver is controversial; however, these data suggest that MKP1 may reduce the p38 and JNK MAPK-mediated transcription of gluconeogenic genes, as well as p38 MAPK-mediated phosphorylation of CREB (cyclic AMP responsive element binding protein), which also promotes gluconeogenesis through PPARγ [154], all of which may lead to increased lipogenesis in the liver. This information, when considered together, demonstrates that MKP1 is upregulated in insulin sensitive tissues in response to a high fat diet in mice and humans. This leads to such metabolic consequences as aberrant glucose metabolism and hepatic steatosis. This demonstrates a key link between obesity, p38 MAPK, immune response and obesity-associated hepatocellular cancer.

## 13. JNK MAPK Pathway in Hepatic Steatosis

The JNK MAPK pathway consists of three proteins—JNK 1, JNK 2 and JNK 3. JNK 1 and 2 are ubiquitously expressed while JNK 3 is restricted to heart, brain and testes [156]. Similar to the p38 MAPK pathway, phosphorylation of the JNK proteins by MAP kinase kinases (specifically MKK4 and 7) and by MAP kinase kinase kinases (MKKK) is required for activation of JNK 1, 2 and 3. Once active, these proteins in turn phosphorylate and activate a wide range of transcription factors and proteins which bring about effects in terms of cell proliferation and differentiation, as well as cell survival and cell death [156]. The role of JNK in cancer seems contradictory [157]. On one hand, JNK signalling is implicated in a wide range of oncogenic pathways including the development of hepatocellular carcinomas [158], lung cancers [159], prostate cancers [160] and brain cancer [161]. However, on the other hand JNK is also known to have tumour suppressor effects [162,163] particularly in melanoma, lymphoma [164] and certain types of skin tumours [165]. These differences seem dependent on the various downstream substrates associated with JNK [157]. The oncogenic function of the JNK pathway depends mostly on its activation of c-JUN, which in turn activates the transcription factor AP1 [162]. This then targets a wide array of genes with AP1 binding sites which regulate the cell cycle, survival and apoptosis. The tumour-suppressive functions of the pathway seem related to their pro apoptotic activity [162].

As JNK3 is not expressed in the liver, studies of the roles of JNK in hepatic metabolism have focused on the JNK1 and JNK2 isoforms. In some studies of obese humans with NASH, hepatic JNK activity has been found to be elevated [166,167]. The enhanced activity of JNK in obese individuals has been attributed to saturated free fatty acids which have been definitively shown to activate JNK pathways [168]. The pro-inflammatory cytokines discussed previously have also been shown to activate JNK through inhibition of MAPK phosphatases by reactive oxygen species [169]. Information on the hepatic functions of JNK also come from animal models with liver specific deficiency of JNK. Deficiency of JNK1 leads to increased insulin resistance [170] whereas deletion of both JNK1 and JNK2 isoforms leads to a state of enhanced insulin and glucose tolerance, increased hepatic insulin action and reduced fasting glucose levels in the setting of a high fat diet [171]. Deletion of JNK2 alone also leads to a similar insulin sensitivity noted with deletion of JNK1 and JNK2 suggesting a key complex role for this isoform in that process.

Hepatic lipid metabolism is also affected by JNK MAPK signalling. Post prandial increases in insulin levels which serve to inhibit the synthesis and storage of lipids are accompanied by a decrease in JNK activity [172]. In mice lacking JNK1 and JNK2, expression of PPARα is activated leading to upregulation of its target genes and increased lipid breakdown. Levels of Fibroblast Growth Factor 21 (FGF21) are increased by PPARα in the liver. In the hyperinsulinemic state in obese individuals, JNK1 and JNK2 activity leads to the inhibition of PPARα, which in turn leads to reduced circulating FGF21, and results in reduced fatty acid oxidation and the deposition of fat in the liver, leading to steatohepatitis.

## 14. ERK MAPK Pathway in Hepatic Steatosis

This pathway is the best understood of the mammalian kinase pathways. Similar to p38 and JNK, ERK 1 and 2 are activated upon phosphorylation by MAP kinase kinases called MKK1 and 2. These, in turn, are activated when phosphorylated by Raf (Raf-1, B-Raf and A-Raf). Raf, in turn, is recruited to the plasma membrane through Ras activity in response to several extracellular signals such as growth factors and cytokines. This pathway has downstream effects on cell proliferation, differentiation, apoptosis and migration via various transcription factors including the AP1 mentioned previously, as well as other factors such as myc.

The precise impact of ERK1/2 signalling in hepatic steatosis is unclear, with conflicting studies in mice as to whether obesity alters hepatic ERK1/2 levels. Moreover, whether ERK1/2 activity is altered in obese humans that progress towards NAFLD or NASH has yet to be established [173]. At present, the role of ERK signalling in NAFLD requires further human studies.

## 15. Conclusions

The role of MAPK signalling in obesity-related cancer remains poorly understood; however, the role of MAPK signalling in immune cells and the regulation of metabolism may be the most important avenues of future research (see Appendix A). Longitudinal human studies, particularly on those undergoing weight loss treatment, will be a key resource for future studies in this area.

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
