# Peer review of "Mitogen-Activated Protein Kinase (MAPK) and Obesity-Related Cancer"

_ijms, 2020, doi:10.3390/ijms21041241_

Round 1

Reviewer 1 Report

In the review manuscript entitled "Mitogen Activated Protein Kinase (MAPK) and obesity related cancer", Fionan Donohoe and colleagues gave a comprehensive overview of how MAPK pathway contributes to obesity related dieseases such as cancer, steatosis, etc. The manuscprit was well written and summarized all obesity related diseases and updated the new knowledge of MAPK pathway in these diseases. I suggest for publication after minor revision.

One suggestion, can authors add one part that include microbiota and obesity, and potential role of MAPK pathway in this? 

Author Response

Thank you for your comments

We have now included a section on the microbiome, MAPK and obesity as suggested

Reviewer 2 Report

IJMS-697250

Mitogen Activated Protein Kinase (MAPK) and obesity related cancer

Several comments for this manuscript:

The structure of abstract is too concise and the objectives are unclear, and needs to be restructured. Line 42-47 description is too brief, should be increased discussion in the relationship between obesity and cancer. Many of the first-ever words should have full names, for example: line 91(pAKT, PTEN), line 95(HbA1c, HOMA-IR,CRP, IL-6), line 153-4 (MAPKK, MAPKKK)…… Lin2 92 and line 106-113, whether the effects of ERK on obesity and cancer are consistent? please confirm. The title of the article does not very fit the text, suggestion whether to modify the title or focus on the title to deeper discussion of the text. Seems to focus primarily on obesity, MAPK and hepatic steatosis. The description is too superficial, need to explain the mechanisms or cause of the two in depth, for example of the paragraph in “5. The impact of obesity on cancer biology” and “6. Obesity related inflammation”.

Author Response

Thank you for your comments

We have updated the abstract and made clearer the uncertain involvement of ERK in this area.

We have also provided full names for abbreviations when first provided as suggested

Round 2

Reviewer 2 Report

None.